**Funding:** Vijayaprakash Suppiah, Elizabeth Hotham and Nerida Packham were co-funded by the

# Acceptability and feasibility of the NPS MedicineWise mobile phone application in supporting medication adherence in patients with chronic heart failure: Protocol for a pilot study

Jessica Chapman-Goetz[1], Nerida Packham[2], Genevieve Gabb[3], Cassandra Potts[4], Kitty Yu[5], Adaire Prosser[4], Elizabeth Hotham[1], Vijayaprakash Suppiah[1,6]*

1 Clinical and Health Sciences, University of South Australia, Adelaide, Australia, 2 Consumer Medicines Information Services, NPS Medicine Wise, Sydney, Australia, 3 Department of Cardiology, Noarlunga GP Plus Super Clinic, Adelaide, Australia, 4 SA Pharmacy, Flinders Medical Centre, Adelaide, Australia, 5 e-Health, NPS Medicine Wise, Melbourne, Australia, 6 Australian Centre for Precision Health, University of South Australia, Adelaide, Australia

* vijay.suppiah@unisa.edu.au

## Abstract

### Introduction

Heart failure (HF) is an increasing global concern. Despite evidence-based pharmacotherapy, morbidity and mortality remain high in HF. Medication non-adherence is a crucial factor in optimising clinical outcomes. A growing number of smartphone applications (apps) assist management. While evidence support their use to promote treatment adherence, apps alone may not be the solution. The objective of this pilot study is to assess the acceptability and feasibility of a tiered intervention added to the NPS MedicineWise dose reminder app (MedicineWise app) in supporting medication adherence in HF.

### Methods and analysis

This prospective, single-blinded, randomised controlled trial will recruit 55 Australian patients with HF to be randomly assigned to either intervention (MedicineWise app + usual care) or control (usual care alone) arm. Control participants will remain unaware of the intervention throughout the study. At baseline, intervention participants will be instructed in the MedicineWise app. A reminder will then prompt medication administration at each dosing interval. If non-adherence is suggested from 24 hourly reports (critical medications) or 72 hours (non-critical medications), the individual/s will be escalated through a tiered, pharmacist-led intervention. The primary outcome will be the acceptability and feasibility of this approach in supporting adherence. Between-group comparison of the Self-Efficacy for Appropriate Medication Use Scale (SEAMS) at baseline, 3 and 6 months will be used to measure the app's value in supporting adherence. Secondary outcome measures include

Innovation Connections Grant scheme by the Department of Industry, Innovation and Science and NPS VentureWise Pty Ltd. Grant no: ICG000775 Funder's website: https://www.industry.gov.au/ Funder's website: https://www.nps.org.au/ The funders had and will not have a role in study design, data collection and analysis, decision to publish, or preparation of the manuscript. Other authors did not receive any direct funding for the work described in this protocol.

**Competing interests:** The authors have declared that no competing interests exist.

self-reported medication adherence and knowledge, health-related quality of life, psychological wellbeing, signs and symptoms of HF, and medication and HF knowledge.

## Ethics and dissemination

The protocol received ethics approval from Central Adelaide Clinical Human Research Ethics Committee (Protocol number R20190302) and University of South Australia Human Research Ethics Committee (Protocol number 202450). Findings will be disseminated through peer-reviewed journals.

## Trial registration number

Australian New Zealand Clinical Trials Registry Clinical trial number: ACTRN12619000289112p (http://www.ANZCTR.org.au/ACTRN12619000289112p.aspx)

## Background

### Chronic heart failure

The term 'heart failure' (HF) refers to cardiac dysfunction whereby blood is pumped at an insufficient volume or rate to meet the body's requirements [1]. Estimated prevalence of HF in Australia is around 1–2%, rising to 10% in those aged 75 years and above [1, 2]. While prevalence is low in comparison with diabetes and cancer, significant morbidity contributes to frequent and prolonged hospitalisations [1, 2]. Reports indicate that approximately four-fifths of HF patients will be hospitalised at least once and that up to three-quarters will die within the first five years post-diagnosis [1, 3]. The burden on the healthcare system accounts for 1–3% of overall healthcare spending [1].

Pharmacotherapy is the foundation of HF management with efficacy across disease severity [4, 5]. Several classes of medication, including Angiotensin Converting Enzyme inhibitors (ACE inhibitors), angiotensin II receptor blockers (ARBs), beta-blockers, diuretics, aldosterone antagonists and digoxin, are effective but clinical outcomes remain suboptimal [3, 4]. Medication non-adherence can increase both the frequency of hospitalisation and the relative risk of all-cause mortality [3].

### The issue of medication non-adherence

Adherence - the extent to which patients follow healthcare professionals' instructions - is, justifiably, a focus of researchers, healthcare providers and healthcare systems [4]. Whatever the condition, regimen complexity or assessment method, the estimated average adherence to long-term pharmacotherapies in developed nations is around 50% [6–8]. The substantial adverse impact of medication non-adherence on individuals, the population and the economy makes this issue a global priority [9, 10].

### Medication-adherence interventions delivered via mobile health

The exponential growth in mobile/smartphone ownership has driven interest in mobile health (mHealth) as a strategy to address non-adherence [9]. In 2020, a staggering 5.8 billion devices were in use globally, giving mHealth interventions the potential to powerfully influence behaviour [11]. Ease of use, accessibility and low-cost make mHealth a desirable modality for

patient-centred services [7, 9, 12]. While the younger generation may be more receptive to mHealth interventions, researchers have determined that older individuals are increasingly incorporating technology into their daily lives [7, 13].

Text message-based interventions provide a simple, acceptable, and feasible platform to promote medication-taking behaviour and research confirms a positive impact [3, 9, 14, 15]. mHealth apps appear similarly effective and may be particularly useful for those with chronic conditions like asthma, diabetes, and heart disease [14–16].

However, technical problems can contribute to high attrition rates and short-lived engagement with mHealth interventions [17, 18]. Without health professional supervision, manual data entry requirements may impede successful, long-term use of dose reminder apps [18]. Further, poor usability could amplify, not alleviate, medication burden [17]. Security and privacy concerns may also deter use [18]. Nevertheless, key organisations, such as the Heart Foundation of Australia and the American Heart Association recognise the promise of mHealth, including dose reminder apps [9, 19, 20]. Therefore, the value in supporting treatment adherence should be assessed in future research [18].

## Research aims

The primary aim of this trial is to assess the acceptability and feasibility of a tiered intervention that utilises the MedicineWise dose reminder app in supporting medication adherence for the treatment of HF.

The secondary aims of this trial are to:

1. determine whether the intervention is associated with enhanced medication and HF knowledge.

2. determine the impact of the smartphone app on signs and symptoms of HF and quality of life at 6 months.

## Methods and analysis

### Study design and participants

This is a prospective, single-blinded randomised controlled trial (RCT) with participants allocated to the intervention or control arm (see Figs 1 & 2). This trial has been registered with the Australian New Zealand Clinical Trials Registry Clinical trial number: ACTRN12619000289112p (http://www.ANZCTR.org.au/ACTRN12619000289112p.aspx )

SEAMS: Self-Efficacy for Appropriate Medication use Scale; SF-36v2: Short Form 36 Health Survey version 2; EQ-5D-5L: instrument name (not an acronym); DASS-21: Depression, Anxiety and Stress Scale-21; SCHFI: Self-Care of Heart Failure Index

Allocation will occur via random number generation, with an even number denoting intervention and an odd number denoting control. Control participants will not be informed of the intervention and will receive a separate consent form to ensure blinding. This is intended to avoid the limitations of prior research with no blinding of controls [21] and strengthen the association between any changes in adherence detected in intervention participants with the use of the reminder app. Researchers undertaking the data collection will not be blinded due to the personalised and tiered nature of the intervention (see Fig 3).

Older individuals or those with comorbidities can be under-represented in mHealth research. [4, 13] To avoid this limitation, potential study participants will not be excluded based on age (unless below 18 years) or coexisting medical conditions (see Table 1). However,

| TIMEPOINT | STUDY PERIOD | | | | | |
| --- | --- | --- | --- | --- | --- | --- |
| | Enrolment | Allocation | Post-allocation | | | Close-out |
| | Pre-baseline | Pre-baseline | Baseline | 3 months | 6 months | Post-study |
| **ENROLMENT:** | | | | | | |
| **Eligibility screen** | X | | | | | |
| **Informed consent** | X | | | | | |
| **Randomisation** | | X | | | | |
| **INTERVENTIONS:** | | | | | | |
| *Tiered intervention + usual care* | | | ←――――――――――――→ | | | |
| *Usual care alone* | | | ←――――――――――――→ | | | |
| **ASSESSMENTS:** | | | | | | |
| *Demographic information* | | | X | | | |
| *Medical history* | | | X | | | |
| *Laboratory test results* | | | X | X | X | |
| *Self-Efficacy for Appropriate Medication Use Scale survey* | | | X | X | X | |
| *Short Form 36 Health Survey version 2* | | | X | X | X | |
| *EQ-5D-5L survey* | | | X | X | X | |
| *Depression Anxiety and Stress Scales survey* | | | X | X | X | |
| *Self-care of HF index survey* | | | X | X | X | |
| *Medication adherence and knowledge survey* | | | X | X | X | |
| *Satisfaction survey (intervention group only)* | | | | | | X |
| *Prescription refill rates & healthcare utilization data collection* | | | | | | X |
| *Cost-effectiveness analysis* | | | | | | X |

**Fig 1. Schedule of enrolment, interventions, and assessments.**

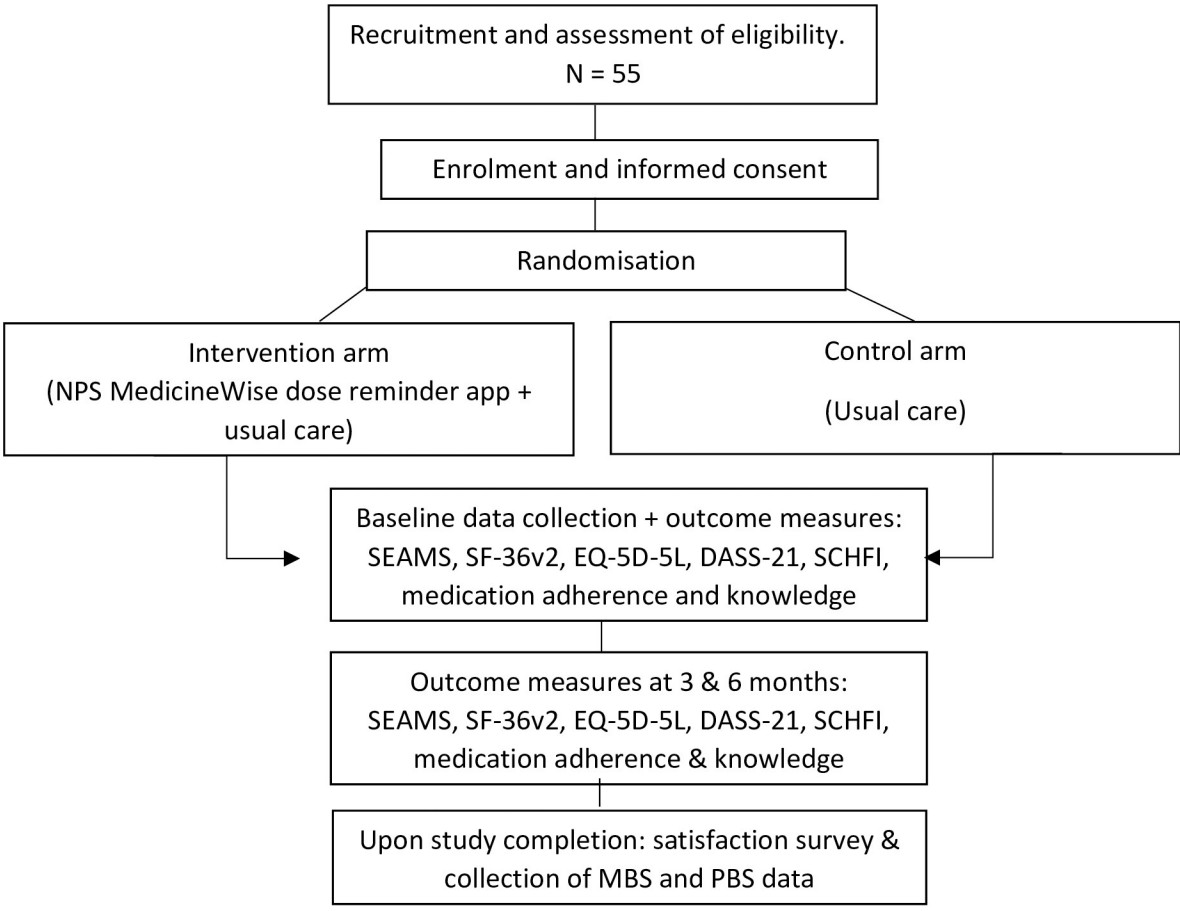

SEAMS: Self-Efficacy for Appropriate Medication use Scale; SF-36v2: Short Form 36 Health Survey version 2; EQ-5D-5L: instrument name (not an acronym); DASS-21: Depression, Anxiety and Stress Scale-21; SCHFI: Self-Care of Heart Failure Index

**Fig 2. Schematic diagram of trial design.**

to optimise study completion, participants should not have palliative HF, severe (NYHA class IV) HF, a life expectancy of less than 6 months, or no access to a smartphone and email.

## Recruitment and sample size

The study will primarily be conducted at five South Australian investigator centres, including the Royal Adelaide Hospital, The Queen Elizabeth Hospital, Lyell McEwin Hospital, Flinders Medical Centre, and Noarlunga Hospital. Rural participants will be recruited through the Integrated Cardiovascular Clinical Network (ICCNet). Eligible participants will be briefed about the study by the treating physician or cardiology nurse either in person or by phone during routine clinic visits or, for rural patients, during their scheduled tele-health consultation/ appointments. Potential participants will be provided with a study summary and referred for eligibility screening (see Table 1) prior to a preliminary interview - face-to-face, at any participating hospital, or via phone. At this initial visit, written informed consent will be obtained from those who agree to participate, and a baseline assessment completed.

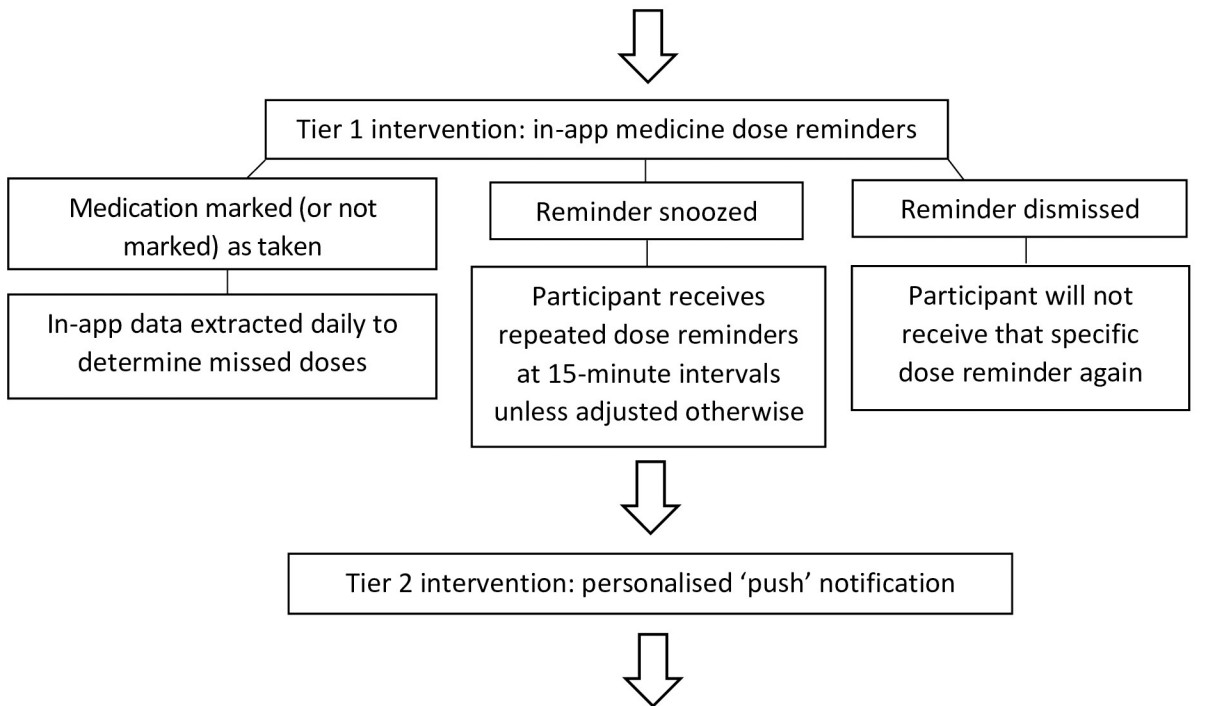

**Fig 3. Schematic process of trial intervention.**

Alternately, interstate participants will be reached via an NPS social media advertisement. A link within the post will direct interested individuals to an eligibility survey. Upon its completion, a preliminary interview will occur as above. Written informed consent will then be obtained via post or email and a baseline assessment completed.

A record of non-participation will be kept for those who were ineligible or who declined involvement. During the consent process it will be emphasised that participants are free to withdraw from the study at any point without impacting their on-going care. If a participant chooses to withdraw, the reason for withdrawal will be requested and recorded.

**Table 1. Eligibility criteria.**

| Inclusion criteria | Exclusion criteria |
|---|---|
| Age 18 years or older | Palliative heart failure |
| Systolic heart failure [+] | NYHA functional class IV |
| NYHA functional class I-III for $\geq$ 3 months | Malignancy or diastolic heart failure |
| LVEF < 50% [*] | Life expectancy $\leq$ 6 months |
| Stable or stabilised condition | Use of other medication reminder apps [**] |
| Participant or carer with access to a smartphone | Unable to read or speak English |
| Able to receive/respond to emails and Skype calls | No access to a smartphone or email |

[+] as confirmed on echocardiography, NYHA: New York Heart Association, LVEF: Left ventricular ejection fraction

[*] at the time of HF diagnosis as per the 2018 Australian HF guidelines

[**] or other electronic reminder systems for daily medication administration.

In the context of a pilot study, sample size calculations were based upon a medium ($d$ = 0.5) to large ($d$ = 0.8) effect size as measured by Cohen's $d$. [22]. While the upper limit of real-world adherence to cardiovascular medications is estimated to be 60% [8], a patient is considered to have good adherence when $\geq$80% of doses are taken [23]. To detect an increase in adherence from 60% to 80%, a minimum total sample size of 52 participants is needed to provide 80% power, with a one tail 0.05 significance level with an effect size of 0.7 [24]. Despite the exclusion of patients undergoing palliative care services or with a life expectancy of less than 6 months, loss to follow-up is likely as the demographic of patients with HF includes the elderly and individuals with comorbidities [25]. Therefore, a total of 55 participants will be recruited between September 2019 and April 2020.

## Tiered interventions

The intervention was developed based on the principle of cued actions as outlined in the Health Belief Model [26]. This model states that external cues, such as reminders and prompts, may increase behaviours that promote adherence. As research suggests, dose reminder apps are not the only solution to address the complex issue of non-adherence [15]. Therefore, this study will utilise a tiered intervention to encourage medication-taking behaviour through a multi-modal approach.

**Control arm.** Participants randomised to the control arm will receive "usual care" for the treatment of HF with medical management, including pharmacotherapy and lifestyle advice, determined by the treating physicians or nurses. To be eligible for inclusion, control participants must not be using any of the smartphone dose reminder apps available for download. This will be confirmed at baseline and again at each follow-up within part A of the SCHFI questionnaire. Additionally, control participants will not have access to the MedicineWise app and will remain blinded to the existence of an intervention arm for study duration. This will be facilitated by using a different patient information sheet and consent form for the recruitment of control and intervention arm participants. These forms for the control arm participants will not mention the existence of the MedicineWise app. Instead, participation in the control arm will involve regular follow ups of functional assessments (questionnaires), medication adherence and knowledge questionnaire and the collection of laboratory results (Fig 2).

**Intervention arm.** In addition to "usual care", participants assigned to the intervention arm will be taught to use the MedicineWise app (see Fig 3). Training will occur either face-to-face or remotely via phone and will begin with downloading the app from the iOS or Android app store. Once an account is registered, the participant will be guided to login and create a

profile. Next, instruction will be given on how to enter all current medications (both HF related and non-related) into the app. Cross-checking with original packaging or a recent medication list will ensure the accurate selection of medication name, strength, dosage form, dose, and frequency of administration. Before each item is saved, the alarm icon will be activated and set to provide a dosing reminder at a time that corresponds with the prescribed regimen. The participant will then be informed on how to edit or delete a medication should their treatment change during the trial. Emphasis will be placed on the importance of maintaining an up-to-date medication profile to avoid incorrect dose reminders or unnecessary interventions. Following this, the participant will be educated on the tiered intervention (see below). At the completion of training, each participant will be asked to demonstrate that they can comfortably perform the required functions. Customer support information within the app will be highlighted and a user manual (developed by the NPS MedicineWise research team) appropriate for either an iOS or Android device will be emailed for ongoing reference.

**Tier 1 intervention: In-app medicine dose reminders from the MedicineWise app.** In-app reminders will be received as notifications and optional audible alarms to prompt administration of each medication dose listed in their profiles. Participants can either: (1) acknowledge a dose has been taken by tapping the orange "TAKE" icon for each medication, (2) "snooze" whereby the notification/ alarm will be repeated twice, at 15-minute intervals or (3) "dismiss" and not receive that specific dose alert again (see Fig 3).

Education will be given to tap the "TAKE" icon as soon as possible following administration to enhance data accuracy. Should a dose be taken, but not immediately recorded, the icon will remain active for tapping until the following day. If a dose is not registered as taken within 24 hours, the app will consider it missed. Information on doses, as logged in the app, will be extracted daily by an NPS MedicineWise research team member to determine if participants missed medication and require escalation to a Tier 2 intervention.

**Tier 2 intervention: Personalised 'push' notification messages from the MedicineWise application.** In collaboration with the NPS MedicineWise researchers and following consultation with a cardiologist and three cardiology pharmacists, medications have been categorised as either *critical* or *non-critical* in the management of HF (see online Table 1 in S3 File). Critical medications are foundational pharmacotherapy due to established morbidity or mortality benefits [27].

If an intervention participant misses a dose of a critical medication for 24 hours, or a non-critical medication for 72 hours, they will receive a personalised 'push' notification within the app. The 24 hour cut-off for receiving the Tier 2 intervention for critical medications is based on evidence of rapid clinical deterioration in HF patients non-adherent for a mere 48 hours [11]. The Tier 2 push notification will remind participants of the importance of adhering to their regimen and to seek help if having difficulty with their medication and/or the app. An NPS MedicineWise research team member will generate a report from the preceding 24 hours and progress participants to the Tier 3 intervention, if required.

**Tier 3 intervention: Phone call intervention delivered by NPS Medicines Line pharmacists.** This tier incorporates use of the NPS Medicines Line - a 'real world' service that operates Mondays to Fridays 9 am to 5 pm AEST (excluding New South Wales public holidays). Participants who do not respond despite two consecutive Tier 2 push notifications will be telephoned by a MedicineWise Medicines Line pharmacist to discuss their medications/adherence (see Fig 3). An NPS MedicineWise researcher will provide the Medicines Line pharmacist with the participant's details and records in a secure file via a shared drive with password access. Only the NPS MedicineWise researchers and Medicines Line pharmacist/s will have access to this file. Details of the call (including the participant's name, date of birth, keywords, and call category) will be recorded in MiDatabank® software where data will be stored, shared

securely and extracted for analysis. A participant can be contacted a maximum of three times by a Medicines Line pharmacist during the trial to discuss adherence.

If outside office hours, participants will receive another 'push' notification until a call can be made the following Monday morning. If participants are concerned about their medications during the weekend, they will be advised to follow their standard care plan and contact their doctor or community pharmacist or present to an emergency department, if required.

## Outcome measures

The primary outcome will be the acceptability and feasibility of the MedicineWise dose reminder app in a tiered, pharmacist-led intervention to support medication adherence for the treatment of HF. Insight into administrative and technical practicalities will be provided by the NPS MedicineWise researchers. Intervention participant perspectives will be obtained via a satisfaction survey at study completion. Comparison of medication adherence between the two arms at baseline, 3 and 6 months, as measured by the self-reported medication adherence tool Self-Efficacy for Appropriate Medication Use Scale (SEAMS), will give some measure of the value of the app in supporting adherence. This tool has been validated previously in other cardiac conditions [28]. Self-reported data will be verified for intervention participants with adherence data captured by the MedicineWise app.

Secondary outcome measures will be collected at baseline, 3 and 6 months. These will include questionnaires on self-reported medication adherence and knowledge (**secondary aim 1**), health-related quality of life (EQ-5D-5L, SF-36v2), psychological wellbeing (Depression Anxiety and Stress Scales) and self-care of HF index (SCHFI) (**secondary aim 2**). Physical signs and symptoms of HF will be captured by the EQ-5D-5L and SF-36v2. The EQ-5D-5L is a subjective measure for evaluating 5 aspects of health that influence quality of life including mobility, self-care, usual activities, pain or discomfort, and anxiety or depression. The SF-36v2 measures 8 health domains, in particular general health, physical and social functioning, role limitations from physical and emotional health problems, bodily pain and vitality.

In the last phase of the study, data on prescription refill rates and healthcare utilisation will be gathered through data linkage with the Australian Pharmaceutical Benefits Scheme (PBS) and Medicare Benefits Schedule (MBS) for the period of 6 months prior to the start of this study till 4 months after completion of this study. The economic evaluation will include direct and non-direct costs of hospitalisations, use of healthcare resources (including medication expenses, GP visits) and death. Researchers will be able to differentiate events that are related to HF as opposed to those not related to HF during the analyses.

The cost-effectiveness of the proposed intervention will be estimated using a cost-utility analysis (CUA) undertaken alongside the RCT with quality adjusted life years gained (QALYs) as measured by the EQ-5D-5L. The purpose of the economic evaluation is to compare any differences in outcomes and resource use that attributable to the intervention. This will be expressed in terms of the incremental cost per QALY gained.

A health system perspective will be gained by economic evaluation including health related resource use (costs and cost-offsets) and health related quality of life outcomes. This evaluation will be conducted alongside the clinical trial for a period of over 16 months for each participant, discounting of costs and benefits beyond the base year of the economic evaluation will not be undertaken.

## Data collection

Baseline data will be collected following written informed consent (see Table 2). All patient data will be collected via face-to-face or phone interviews according to participant preference,

**Table 2. Baseline data.**

| Category | Example |
|---|---|
| Sociodemographic information | Date of birth, gender, ethnicity, height, weight, education/marital/employment/ smoking status |
| Medical history | Comorbidities, NYHA class, all currently prescribed and non-prescribed medications |
| Laboratory test results | Full blood count, electrolytes (including sodium, potassium, creatinine +/- magnesium), Hb, iron studies, NT pro BNP, liver function tests |
| Questionnaires | Self-Efficacy for Appropriate Medication Use Scale (SEAMS); Short Form 36 Health Survey version 2 (SF-36v2); EQ-5D-5L; Depression Anxiety and Stress Scales (DASS-21); Self-Care of Heart Failure Index (SCHFI); medication adherence and knowledge questionnaires |

geographical location, or SA Health guidelines for COVID-19 infection control. At each follow up, all questionnaires will be repeated and details of any changes to participants' medication or intervening hospital admissions will be recorded.

To encourage participant retention and completeness of data collection, participants will receive a honourarium of AUD$50 at the 6-month data collection point.

PBS and MBS utilisation data from 1 February 2019 to 31 January 2021 will be obtained from the Department of Human Services following informed patient (or proxy in the case of moderate to severe cognitive impairment) consent. Unit costs will be derived from relevant hospital finance departments, published data sets including Pharmaceutical Benefits Scheme and Medicare Benefits Schedule and Australian Refined Diagnosis Related Groups (AR-DRG) cost weights.

## Analysis

Thematic analysis will be carried out of participant responses to open-ended questions in the functional assessments and satisfaction survey to generate emerging and overarching themes. This analysis will characterise intervention arm participants' observed and self-reported responses to the intervention collected at months 3 and 6, and link these responses to the acceptability, utility, and their engagement with the MedicineWise app.

Between-group (intervention vs. control arm) quantitative analysis will occur at the different time points. Continuous and discreet variables will be analysed through SPSS software using independent t-tests and chi-squared tests, respectively. Non-parametric statistical tests such as Mann Whitney U tests will be employed according to the nature of the data. All between-group findings that are significant ($\alpha$ 0.05) will be reported with odds ratios and 95% confidence intervals.

A participant level analysis will be undertaken to determine the incremental costs and outcomes associated with the intervention relative to controls. Incremental cost effectiveness ratios and their associated confidence intervals will be estimated and cost effectiveness acceptability curves for varying threshold values of cost effectiveness will also be presented. An assessment of the sensitivity of the results obtained to variation in measured resource use, effectiveness and/or unit costs will be undertaken using appropriate one-way, multi-way and probabilistic sensitivity analysis.

## Patient and public involvement

There has been no specific patient or public involvement in the design or planning of this study.

## Ethics and dissemination

The study will be conducted in accordance with the ethical principles of the Declaration of Helsinki and in adherence with the National Health and Medical Research Council's National Statement on Ethical Conduct in Human Research. Ethics approval was obtained from the Central Adelaide Clinical Human Research Ethics Committee (Protocol number R20190302) and the University of South Australia Human Research Ethics Committee (Protocol number 202450) in 8th April 2019 and 22nd July 2019 respectively.

Participants' names, addresses and smart phone numbers will be collected for the purpose of intervention delivery and participant interviews. All data collected from participants will otherwise be stored electronically as de-identified data separately from identifiable data. All paper-based data will be stored in secure cabinets with restricted access. Electronic records will be stored in a secure server maintained by the University of South Australia. Access to all data will be restricted to the members of the research team and will not be shared with any third parties.

A manuscript based upon the study results will be developed and submitted to peer-reviewed journals for publication. Findings will also be incorporated into a thesis for submission and examination as part of Master's by Research in Pharmacy at the University of South Australia.

## Supporting information

**S1 File. Medicine adherence in chronic heart failure.**
(DOCX)

**S2 File. SPIRIT 2013 checklist: Recommended items to address in a clinical trial protocol and related documents***.
(DOC)

**S3 File.**
(DOCX)

## Acknowledgments

The authors would like to thank all participants, cardiologists and cardiology nurses for their involvement and support throughout the study.

## Author Contributions

**Conceptualization:** Nerida Packham, Elizabeth Hotham, Vijayaprakash Suppiah.

**Data curation:** Jessica Chapman-Goetz.

**Formal analysis:** Jessica Chapman-Goetz, Elizabeth Hotham, Vijayaprakash Suppiah.

**Funding acquisition:** Nerida Packham, Elizabeth Hotham, Vijayaprakash Suppiah.

**Investigation:** Jessica Chapman-Goetz.

**Methodology:** Jessica Chapman-Goetz, Nerida Packham, Genevieve Gabb, Cassandra Potts, Kitty Yu, Adaire Prosser, Elizabeth Hotham, Vijayaprakash Suppiah.

**Project administration:** Jessica Chapman-Goetz, Nerida Packham, Vijayaprakash Suppiah.

**Supervision:** Nerida Packham, Genevieve Gabb, Kitty Yu, Elizabeth Hotham, Vijayaprakash Suppiah.

**Writing – original draft:** Jessica Chapman-Goetz, Nerida Packham, Cassandra Potts, Kitty Yu, Adaire Prosser, Elizabeth Hotham, Vijayaprakash Suppiah.

**Writing – review & editing:** Nerida Packham, Genevieve Gabb, Kitty Yu, Elizabeth Hotham, Vijayaprakash Suppiah.

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
