## [Decision Letter · Decision Letter 0]

27 Sep 2021

PONE-D-21-20275

Acceptability and feasibility of the NPS MedicineWise mobile phone application in supporting medication adherence in patients with chronic heart failure: Protocol for a pilot study

PLOS ONE

Dear Dr. Suppiah,

Thank you for submitting your manuscript to PLOS ONE. After careful consideration, we have decided that your manuscript does not meet our criteria for publication and must therefore be rejected.

I am sorry that we cannot be more positive on this occasion, but hope that you appreciate the reasons for this decision.

Yours sincerely,

Yoshihiro Fukumoto

Academic Editor

PLOS ONE

Reviewers' comments:

Reviewer's Responses to Questions

**Comments to the Author**

1. Does the manuscript provide a valid rationale for the proposed study, with clearly identified and justified research questions?

Reviewer #1: Yes

Reviewer #2: No

2. Is the protocol technically sound and planned in a manner that will lead to a meaningful outcome and allow testing the stated hypotheses?

Reviewer #1: Yes

Reviewer #2: No

3. Is the methodology feasible and described in sufficient detail to allow the work to be replicable?

Reviewer #1: Yes

Reviewer #2: No

4. Have the authors described where all data underlying the findings will be made available when the study is complete?

Reviewer #1: No

Reviewer #2: No

5. Is the manuscript presented in an intelligible fashion and written in standard English?

Reviewer #1: Yes

Reviewer #2: Yes

6. Review Comments to the Author

You may also provide optional suggestions and comments to authors that they might find helpful in planning their study.

Reviewer #1: This is a study protocol to evaluate the acceptability and feasibility of the NPS MedicineWise mobile phone application in

supporting medication adherence in patients with chronic heart failure. This E-health related trial looks interesting. I have several comments to the authors.

1. Blind of intervention: The authors stated control participants will not have access to the NPS MedicineWise app and will remain "blinded" to the existence of an intervention arm for study duration. But when enrollment, the authors will explain what is the intervention or control in this study. So, I think the participants would know that they are going to control arm at randomization.

2. Why the authors will exclude HF patients with preserved ejection fraction? Is this because there are still limited data in medication what improve prognosis in HFpEF?

3. Secondary outcome will be signs and symptoms of HF but Table 2 looks like the authors will collect NYHA class only.

4. If I understand correctly, most data will be obtained by questionnaire and missing assessment by attending physicians.

Reviewer #2: This manuscript is a design paper of a clinical study which will analyze the usefulness of smartphone applications assist management. This is a prospective, single-blinded, randomized controlled trial in 55 Australian patients with heart failure. Although the authors analyze the NYHA class, laboratory test including NT-proBNP, and questionnaires at 6 months, the clinical events such as hospitalization, and cardiac death, were not analyzed in this study. Therefore, the effect of this smartphone application for cardiac events can not be elucidated by this clinical study.

7. PLOS authors have the option to publish the peer review history of their article (what does this mean?). If published, this will include your full peer review and any attached files.

Reviewer #1: No

Reviewer #2: No

- - - - -

---

## [Author Response · Author response to Decision Letter 0]

16 Nov 2021

Reviewers’ comments

Reviewer #1: This is a study protocol to evaluate the acceptability and feasibility of the NPS MedicineWise mobile phone application in supporting medication adherence in patients with chronic heart failure. This E-health related trial looks interesting. I have several comments to the authors.

1. Blind of intervention: The authors stated control participants will not have access to the NPS MedicineWise app and will remain "blinded" to the existence of an intervention arm for study duration. But when enrolment, the authors will explain what is the intervention or control in this study. So, I think the participants would know that they are going to control arm at randomization.

On page 6 of the manuscript (Para 3; Heading: Control arm), we have amended the section on the recruitment of control arm participants as follows. The amended sections are in italics. We hope that we have adequately addressed this reviewer’s concern about the blinding of the control arm participants. 

“Participants randomised to the control arm will receive “usual care” for the treatment of HF with medical management, including pharmacotherapy and lifestyle advice, determined by the treating physicians or nurses. To be eligible for inclusion, control participants must not be using any of the smartphone dose reminder apps available for download. This will be confirmed at baseline and again at each follow-up within part A of the SCHFI questionnaire. Additionally, control participants will not have access to the MedicineWise app and will remain blinded to the existence of an intervention arm for study duration. This will be facilitated by using a different patient information sheet and consent form for the recruitment of control and intervention arm participants. These forms for the control arm participants will not mention the existence of the MedicineWise app. Instead, participation in the control arm will involve regular follow ups of functional assessments (questionnaires), medication adherence and knowledge questionnaire and the collection of laboratory results (Figure 2).”

2. Why the authors will exclude HF patients with preserved ejection fraction? Is this because there are still limited data in medication what improve prognosis in HFpEF?

This decision was based on the medical advice received from our cardiologist collaborators. 

3. Secondary outcome will be signs and symptoms of HF but Table 2 looks like the authors will collect NYHA class only.

In addition to the NYHA class, we are also interested in the physical signs and symptoms which will be captured by the EQ-5D-5L and SF-36v2. The EQ-5D-5L is a subjective measure for evaluating 5 aspects of health that influence quality of life including mobility, self-care, usual activities, pain or discomfort, and anxiety or depression. The SF-36v2 measures 8 health domains, in particular general health, physical and social functioning, role limitations from physical and emotional health problems, bodily pain and vitality. We believe that these aspects of the two questionnaires would have adequately capture physical signs and symptoms of HF. 

For clarity, we have added the following sentences to the bottom of the second paragraph in the outcome section (Page 8): 

“Physical signs and symptoms of HF will be captured by the EQ-5D-5L and SF-36v2. The EQ-5D-5L is a subjective measure for evaluating 5 aspects of health that influence quality of life including mobility, self-care, usual activities, pain or discomfort, and anxiety or depression. The SF-36v2 measures 8 health domains, in particular general health, physical and social functioning, role limitations from physical and emotional health problems, bodily pain and vitality.”

4. If I understand correctly, most data will be obtained by questionnaire and missing assessment by attending physicians.

This is a pilot study that aims to test the acceptability and feasibility of the NPS MedicineWise app in supporting medication adherence in HF patients. The intervention that we are attempting to trial is the tiered intervention supported by pharmacists. Including the assessment of physicians is beyond the scope of this study. Once we have shown that this tiered intervention is feasible from the view of the service provider (NPS MedicineWise ltd) and is widely accepted by HF patients, we can then go on to do a much larger scale study in which we can incorporate other measures such as assessments from attending physicians, which will also include further clinician reported signs and symptoms of HF. 

Reviewer #2: This manuscript is a design paper of a clinical study which will analyze the usefulness of smartphone applications assist management. This is a prospective, single-blinded, randomized controlled trial in 55 Australian patients with heart failure. Although the authors analyze the NYHA class, laboratory test including NT-proBNP, and questionnaires at 6 months, the clinical events such as hospitalization, and cardiac death, were not analyzed in this study. Therefore, the effect of this smartphone application for cardiac events can not be elucidated by this clinical study.

Last paragraph on page 8 describes “the cost effectiveness analysis which will include the collection of prescription refill rates and healthcare utilisation will be gathered through data linkage with the Australian Pharmaceutical Benefits Scheme (PBS) and Medicare Benefits Schedule (MBS). The purpose of the economic evaluation is to compare any differences in outcomes and resource use that attributable to the intervention. This will be expressed in terms of the incremental cost per QALY gained. 

A health system perspective will be gained by economic evaluation including health related resource use (costs and cost-offsets) and health related quality of life outcomes. This evaluation will be conducted alongside the clinical trial with a 6 month follow up period, discounting of costs and benefits beyond the base year of the economic evaluation will not be undertaken.”

The economic evaluation will include hospitalisations, use of healthcare resources (including hospitalization) and death. We believe that we have sufficiently addressed this reviewer’s only comment in the manuscript. 

To clarify this information, paras 2, 3 and 5 of the outcome measures section (Page 8) has been amended as follows with the amended sentences in italics.

Secondary outcome measures will be collected at baseline, 3 and 6 months. These will include questionnaires on self-reported medication adherence and knowledge (secondary aim 1), health-related quality of life (EQ-5D-5L, SF-36v2), psychological wellbeing (Depression Anxiety and Stress Scales) and self-care of HF index (SCHFI) (secondary aim 2). Physical signs and symptoms of HF will be captured by the EQ-5D-5L and SF-36v2. The EQ-5D-5L is a subjective measure for evaluating 5 aspects of health that influence quality of life including mobility, self-care, usual activities, pain or discomfort, and anxiety or depression. The SF-36v2 measures 8 health domains, in particular general health, physical and social functioning, role limitations from physical and emotional health problems, bodily pain and vitality.

In the last phase of the study, data on prescription refill rates and healthcare utilisation will be gathered through data linkage with the Australian Pharmaceutical Benefits Scheme (PBS) and Medicare Benefits Schedule (MBS) for the period of 6 months prior to the start of this study till 4 months after completion of this study. The economic evaluation will include direct and non-direct costs of hospitalisations, use of healthcare resources (including medication expenses, GP visits) and death. Researchers will be able to differentiate events that are related to HF as opposed to those not related to HF during the analyses. 

Paragraph 4

A health system perspective will be gained by economic evaluation including health related resource use (costs and cost-offsets) and health related quality of life outcomes. This evaluation will be conducted alongside the clinical trial for a period of over 16 months for each participant, discounting of costs and benefits beyond the base year of the economic evaluation will not be undertaken.

---

## [Decision Letter · Decision Letter 1]

17 Jan 2022

Acceptability and feasibility of the NPS MedicineWise mobile phone application in supporting medication adherence in patients with chronic heart failure: Protocol for a pilot study

PONE-D-21-20275R1

Dear Dr. Suppiah,

We’re pleased to inform you that your manuscript has been judged scientifically suitable for publication and will be formally accepted for publication once it meets all outstanding technical requirements.

Kind regards,

Yoshihiro Fukumoto

Academic Editor

PLOS ONE

Additional Editor Comments (optional):

Reviewers' comments:

Reviewer's Responses to Questions

**Comments to the Author**

1. Does the manuscript provide a valid rationale for the proposed study, with clearly identified and justified research questions?

Reviewer #1: Yes

Reviewer #2: No

Reviewer #3: Yes

2. Is the protocol technically sound and planned in a manner that will lead to a meaningful outcome and allow testing the stated hypotheses?

Reviewer #1: Yes

Reviewer #2: No

Reviewer #3: Yes

3. Is the methodology feasible and described in sufficient detail to allow the work to be replicable?

Reviewer #1: Yes

Reviewer #2: No

Reviewer #3: Yes

4. Have the authors described where all data underlying the findings will be made available when the study is complete?

Reviewer #1: Yes

Reviewer #2: No

Reviewer #3: Yes

5. Is the manuscript presented in an intelligible fashion and written in standard English?

Reviewer #1: Yes

Reviewer #2: Yes

Reviewer #3: Yes

6. Review Comments to the Author

You may also provide optional suggestions and comments to authors that they might find helpful in planning their study.

Reviewer #1: The authors addressed the comments properly and the revised manuscript improved significantly. I do not have further comments to the authors.

Reviewer #2: This reviewer understand that the authors will evaluate the effect of a smartphone application on cost effectiveness and will use the questionnaires such as SF-36v2, EQ-5D-5L, Depression Anxiety and Stress Scales and selfcare of HF index (SCHFI).

Reviewer #3: This is a study protocol to assess the acceptability and feasibility of a tiered intervention added to the NPS MedicineWise dose reminder app (MedicineWise app) in supporting medication adherence in HF. I think it is very interesting. I have no comments.

7. PLOS authors have the option to publish the peer review history of their article (what does this mean?). If published, this will include your full peer review and any attached files.

Reviewer #1: No

Reviewer #2: No

Reviewer #3: No

---

## [Editor Report · Acceptance letter]

25 Jan 2022

PONE-D-21-20275R1 

Acceptability and feasibility of the NPS MedicineWise mobile phone application in supporting medication adherence in patients with chronic heart failure: Protocol for a pilot study 

Dear Dr. Suppiah:

I'm pleased to inform you that your manuscript has been deemed suitable for publication in PLOS ONE. Congratulations! Your manuscript is now with our production department. 

Kind regards, 

on behalf of

Dr. Yoshihiro Fukumoto 

Academic Editor

PLOS ONE